# Breastfeeding in the Community—How Can Partners/Fathers Help? A Systematic Review

**DOI:** 10.3390/ijerph17020413

**Published:** 2020-01-08

**Authors:** Felix Akpojene Ogbo, Blessing J. Akombi, Kedir Y. Ahmed, Abdon G. Rwabilimbo, Akorede O. Ogbo, Noel E. Uwaibi, Osita K. Ezeh, Kingsley E. Agho

**Affiliations:** 1Translational Health Research Institute (THRI), School of Medicine, Western Sydney University, Campbelltown Campus, Locked Bag 1797, Penrith 2571, NSW, Australia; K.Ahmed@westernsydney.edu.au (K.Y.A.); rwabi1977@gmail.com (A.G.R.); ezehosita@yahoo.com (O.K.E.); K.agho@westernsydney.edu.au (K.E.A.); 2General Practice Unit, Prescot Specialist Medical Centre, Welfare Quarters, Makurdi 972261, Benue State, Nigeria; kody.awosemo@gmail.com; 3School of Public Health and Community Medicine, Faculty of Medicine, University of New South Wales, Sydney 2052, NSW, Australia; b.akombi@unsw.edu.au; 4College of Medicine and Health Sciences, Samara University, P.O. Box 132, Samara, Ethiopia; 5Chato District Council, Geita region, Northwestern, Tanzania; 6College of Medicine, Edo University Iyamho, Kilometer 7 Auchi–Abuja expressway, Auchi, Edo State, Nigeria; noel.uwaibi@gmail.com

**Keywords:** partner/father, support, breastfeeding, impact/influence, systematic review

## Abstract

Support from partners/fathers and families can play a significant role in a mother’s decision to initiate, continue or cease breastfeeding postnatally. This study systematically reviewed published studies to determine the impact of specific types of partner support on breastfeeding initiation, duration and exclusivity. We used the 2015 Preferred Reporting Items for Systematic reviews and Meta-Analysis (PRISMA) guidelines for the review. Seven computerized bibliographic databases (Embase, ProQuest Central, Scopus, PsycINFO, Web of Science, MEDLINE/PubMed and CINAHL) were searched. Of a total of 695 articles retrieved from the databases, seven studies met the inclusion criteria and reported on breastfeeding initiation, duration and exclusivity. Four of the seven studies found that partner support in the form of verbal encouragement to new mothers increased breastfeeding duration and exclusivity. Other types of partner supportive actions that led to improved breastfeeding behavior included sensitivity of the partner to the nursing mother’s needs, assistance in preventing and managing breastfeeding difficulties, and helping with household and child care duties. This review showed that specific supportive actions of partners/fathers in the community positively improved breastfeeding practices. To maximise the impact of breastfeeding policies and interventions among new mothers, breastfeeding programmes should consider the involvement of partners/fathers and their specific roles.

## 1. Introduction

Early initiation of breastfeeding (EIBF) and exclusive breastfeeding (EBF) are effective strategies to improve the global child survival rate. EIBF is the introduction of human breastmilk to infants within the first 24 h of birth, while EBF is giving infants aged 0–5 months only human breastmilk (and oral rehydration solution, or drops/syrups of vitamins, minerals, or medicines as required) with no additional food or drink [1]. The short- and long-term benefits of EIBF and EBF are well documented. EIBF reduces the risk of neonatal morbidity and mortality, as it prevents the introduction of contaminated prelacteal foods (water, rice water, herbal mixture or juice), as well as deprives newborns of colostrum—rich in nutrients and immunoglobulins. EBF protects against diarrheal disease (a leading cause of global child deaths) [2,3,4] and childhood obesity [5], and is likely to increase childhood neurocognitive functioning [6]. Improved maternal health outcomes (i.e., protection against breast cancer, improved birth spacing and a reduced risk of developing type 2 diabetes) have been reported in mothers who exclusively breastfed [2].

In 2019, only 41% of infants under six months of age were exclusively breastfed worldwide [7]. This estimate may vary widely in low- and middle-income countries (LMICs) and in high-income countries [2]. For example, studies conducted in Economic Community of West African States (ECOWAS) countries showed that EIBF prevalence ranged from 17% in Guinea to 62% in Togo and Liberia [8]. Similarly, EBF rates among infants 6 months or younger ranged from 13% in Côte d’Ivoire to 58% in Togo among ECOWAS countries [8]. Similar variations in breastfeeding practices have been reported in Australia [9] and European countries [10]. To improve global breastfeeding rates, the World Health Organisation endorsed a set of Global Nutrition Targets (WHO GNT, including Goal–5), which aims to increase the global EBF rate to at least 50% by the year 2025 [11]. However, a recent study indicated that only three African countries (Guinea-Bissau, Rwanda, and São Tomé and Príncipe) are on track to meet the GNT for EBF [12]. 

Past studies from both LMICs and high-income countries have reported wide heterogeneity in the determinants of EBF. For example, research conducted in India had suggested that the determinants of non-EBF included higher maternal education in Southern India and belonging to rich households in Central India. In contrast, the determinants of EBF were higher maternal education in the Central region and frequent antenatal care (≥4) visits in Northern India [13]. In sub-Saharan Africa, countries with high diarrhea mortality, higher maternal education and household wealth were associated with EBF [14,15,16], while maternal employment was also related to EBF in Ethiopia [17]. The important role of grandmothers in reducing the likelihood of EBF has been described in northern Malawi [18], southern Nigeria [19] and internationally [20]. In a high-income country like Australia, a lack of maternal prenatal breastfeeding intention [21], no partner support, perinatal depression and intimate partner violence were associated with non-EBF [22,23]. Anxiety about breastfeeding in public has been reported as an emerging barrier to EBF in Australia and European countries [22,24,25].

Globally, previous research [22,23,26,27,28] has shown that family members (i.e., husband, partner or grandmother) do not only influence a mother’s decision to initiate and continue breastfeeding, but also play a significant role in the premature cessation of appropriate breastfeeding in the early postnatal period. For example, Ogbo et al. found that partner support was associated with appropriate EBF in the first 6 weeks of birth among Australian women [22] and those from culturally and linguistically diverse populations [23]. Similarly, two previous systematic reviews showed that breastfeeding interventions which considered increasing partners/fathers breastfeeding information resulted in improvements in breastfeeding outcomes [29,30]. However, despite increasing opportunities in the workplace for fathers worldwide to support new mothers (including prospects to improve breastfeeding) [31,32], there is often limited attention, if any, to the specific types of the supportive role of partners/fathers in mothers’ decision to initiate, continue or cease breastfeeding in the postnatal period. In addition, most breastfeeding proponents and programme planners often design breastfeeding interventions that target new mothers, with no corresponding and in-depth understanding of who else in the household can significantly influence infant feeding decisions [20,33].

As part of a wider modernization paradigm shift which views women as only primary carers of infants and young children [34,35,36], and gender role differentiation [37,38,39], attention must be paid to the broader household and community contexts in which other actors with hierarchical patterns of authority operate and influence infant feeding [33]. Understanding the specific types of supportive role of partners for new mothers in the context of breastfeeding initiation, duration and exclusivity will be useful to breastfeeding advocates, health practitioners and policy decision makers in the protection, promotion and support of appropriate breastfeeding practices. Accordingly, we aimed to systematically assess the impact of specific types of partner support on breastfeeding initiation, duration and exclusivity.

## 2. Materials and Methods 

This systematic review adhered to the 2015 Preferred Reporting Items for Systematic Reviews and Meta-Analyses (PRISMA) guidelines [40] and was conducted following the four-stage PRISMA flowchart [40,41]. The PRISMA is an evidence-based minimum set of items that aims to support authors to comprehensively report systematic reviews to (i) help clinicians keep up to date; (ii) provide robust evidence for policy decision makers to examine risks, benefits, and harms of health care interventions; (iii) collate and summarize closely related research for patients and their carers; and (iv) provide a starting point for the development of clinical practice and public health guidelines [41]. The PRISMA guidelines focus on what authors can follow to ensure the transparent and complete reporting of systematic reviews and meta-analyses. The PRISMA statement is a checklist which consist of 27 items, including title, abstract, methods, results, discussion and funding to meet reporting standards of systemic reviews and meta-analyses. Additional information on the PRISMA statement has been published elsewhere [40,41].

In this review, partner/father was defined as a husband/partner of the new mother or the father of the infant whose breastfeeding outcome was measured. Breastfeeding initiation was measured as the introduction of breastmilk to the infant within the first 24 h of birth, while duration of breastfeeding was defined as any breastfeeding up to 24 months postpartum. Breastfeeding exclusivity (EBF) was measured based on the provision of only human breastmilk (and oral rehydration solution, or drops/syrups of vitamins, minerals, or medicines as needed) to infants aged 0–5 months [42]. 

### 2.1. Search Strategy

An initial search of the Cochrane library and Google Scholar was performed to ensure no previous study on the impact of specific types of partner support on breastfeeding initiation, duration and exclusivity had been conducted. Thereafter, a list of relevant medical subject headings (MeSH) words and sub-headings of keywords was generated and used to extensively search for peer-reviewed articles from seven computerized bibliographic databases (Embase, ProQuest Central, Scopus, PsycINFO, Web of Science, MEDLINE/PubMed and CINAHL). Search terms were slightly adjusted to suit each database. The articles retrieved from each database were imported into an EndNote library. For additional relevant publications that might have been missed, we searched the bibliographical references of all retrieved articles that met the inclusion criteria, complemented by citation tracking using Google Scholar. The following combination of search terms and keywords was used in the search:

Breastfeed *

AND partner or husband or paternal or spouse or father or dad or male

AND support OR involvement OR assistance OR participation

AND impact or effect OR influence

### 2.2. Inclusion and Exclusion

Studies were included in the review if they (i) were peer-reviewed articles, dissertations, books and book chapters, working papers, technical reports, discussion papers, and conference papers; (ii) measured the specific types of partner supportive actions on at least one of the outcome measures (breastfeeding initiation, duration, or exclusivity); and (iii) were written in English and their full-texts were available and accessible. The search was not restricted by date or location to leverage the extensive global research on infant feeding [43] and the WHO/UNICEF interventions to protect, promote and support breastfeeding [44,45]. A search log was developed and used for accountability and transparency.

Studies were excluded if they (i) were published in languages other than English; (ii) were reviews, editorials, letters to editors, and opinion pieces; and (iii) did not research the impact of the specific types of partner support and breastfeeding initiation, duration and exclusivity.

### 2.3. Data Extraction 

Eligible studies retrieved from the seven computerized bibliographic databases and manual bibliographic searches of full-text articles were imported into an EndNote library and duplicates were removed. The screening of the retrieved studies was conducted on the basis of titles and abstracts to determine relevance. Full-texts of the remaining studies were read for eligibility and the studies that met the inclusion criteria were retained. The process of data extraction and appraisal of retrieved studies was conducted by one author (BJA) and independently reviewed by a second author (KYA). Both authors perused the reference lists of the retained studies to identify additional relevant studies. A third reviewer (FAO) adjudicated the differences that emerged in the selection of the final studies for inclusion.

Piloted forms adapted from the Joanna Briggs Institute (JBI) reviewer’s manual [46] and the Cochrane Handbook for Systematic Reviews [47] were used in the extraction of qualitative and quantitative data. Eligible studies were identified by their author, year of publication, country, study design, study characteristics, types of partner support, impact of partner support and study limitations.

### 2.4. Quality Assessment

Quality assessment involved the analysis of the methodological quality of the retained studies. This informed the evaluation of study limitations and appropriateness of methods in addressing research objectives. It highlighted key concepts for evaluating the internal validity of the selected studies by considering the potential risk of selection bias, measurement bias, or confounding. The quality of the eligible studies was assessed using the study assessment tools of the National Heart, Lung, and Blood Institute of the National Institutes of Health (NIH) for quality assessment of Observational Cohort and Cross-Sectional Studies and Controlled Intervention Studies [48]. The NIH checklist for each study type measures 14 unique criteria to assess the internal validity of studies. Studies were considered as ‘*good*’ if they met 10–14 criteria, ‘*fair*’ if they met 5–9 criteria and ‘*poor*’ if they met ≤4 criteria as shown in Appendix A. A high-quality rating implies a low risk of bias and vice versa [48]. Emerging evidence has suggested that the NIH checklist is a robust tool for assessing risk of bias in observational and experimental studies [49,50,51].

By systematically reviewing previously published studies, this study did not require ethical approval.

## 3. Results

A total of 695 articles were retrieved from seven databases, and a manual search of the bibliographic references of the full-text articles yielded an additional two articles. After the removal of duplicates, 659 articles were retained. A screening of the titles and abstracts resulted in the exclusion of 613 articles. The full-text of the remaining 44 articles were reviewed, and then 37 articles were further excluded as they did not meet the full inclusion criteria, and seven articles met the inclusion criteria as shown in Figure 1. In this review, only published peer-reviewed articles met the full criteria for inclusion in this study, and any other forms of studies (dissertations, books and book chapters, working papers, technical reports, discussion papers, and conference papers) were not found during the search.

### 3.1. Characteristics of Included Studies

Table 1 shows a summary of eligible studies in this review. Study designs employed across eligible studies include controlled clinical trials (2), a prospective cohort design (1), cross-sectional questionnaire-based studies (3), and a qualitative exploratory design (1). Sample size ranged from 34 couples to 1174 mothers. The mean age of participants was >20 years. Eligible studies were conducted between 2002 and 2017.

### 3.2. Evidence from Reviewed Studies

The impact of the specific types of partner support on breastfeeding initiation, duration and exclusivity varied in the eligible studies. Three studies measured mothers’ perception or the types of supportive actions received from their partners [52,54,56], while the other studies measured the types of support provided to the new mother by her partner, or the perception of the couple of what constituted paternal support for breastfeeding [53,55,57,58,59].

In this review, one study measured breastfeeding duration for up to 6 months [55], and another study measured breastfeeding duration at 1, 3, 6, 9, 12, 18 and 24 months post birth [53]. Only one eligible study measured breastfeeding initiation, duration and exclusivity at 6 weeks and 6 months postnatally [56], while one study measured EBF for up to 6 weeks postpartum [54]. One study measured breastfeeding duration for at least 6 months and more than 6 months after mothers returned to work [52]. One study measured EBF at 6 months and breastfeeding duration for up to 12 months after birth [57], while another study measured EBF for at least 6 months and breastfeeding duration for up to the time of weaning (6–8 months) [58].

Of the seven eligible studies, four reviewed studies reported that partner supportive action in the form of providing encouragement to the new mother was a key strategy to improve breastfeeding practices [52,54,55,56]. Tsai [52] showed that partner’s words of encouragement to use the lactation rooms (OR = 6.57; 95% CI: 4.21, 10.4) or take milk expression breaks (OR = 2.84; 95% CI: 1.62, 5.23) increased breastfeeding duration for the first 6 months or more after returning to work. Ingram et al. [54] found that words of encouragement from partners was strongly associated with the maintenance of EBF in the first 6 weeks postpartum (OR = 3.25; 95% CI: 1.95, 5.42). Similarly, Tarrant et al. [56] found that partner support in the form of positive encouragement to mothers improved breastfeeding initiation, duration and exclusivity at 6 weeks and 6 months postpartum.

Only one study specifically investigated the relationship between the types of partner support and breastfeeding duration. Rempel et al. [53] conducted two studies to assessed the relationships between fathers’ breastfeeding support, mothers’ perceptions of the support received with breastfeeding satisfaction and duration. Study 1 showed that responsiveness of fathers (fathers’ sensitivity to the mothers’ needs and respect for her decisions) resulted in longer duration of breastfeeding. In study 2, however, the appreciation of fathers (behaviors of encouragement and valuing the breastfeeding mother) led to shorter duration of breastfeeding. In both studies, mothers’ perceptions of their partners’ responsiveness predicted longer breastfeeding duration.

Of the seven studies included in this review, there was only one study that used a qualitative exploratory approach. Tohotoa et al. [55] examined couples’ perceptions of what constituted support for breastfeeding, with a focus on paternal support. The study conducted focus group discussions and interviews with mothers and an online survey for fathers. The authors found that partners’ words of encouragement to mothers “*to do your [their] best*” and offering mothers acknowledgement for the breastfeeding effort and giving emotional support increased breastfeeding duration. The study also reported that partners who anticipated new mothers’ needs and got the job done, and partners who demonstrated commitment to breastfeeding also led to improved breastfeeding practice.

Only two of the seven studies involved controlled clinical trials. Pisacane et al. [57] reported that partner supportive actions in the form of assisting with preventing and managing lactation difficulties resulted in increased rates of breastfeeding duration and exclusivity. In the second trial, Susin et al. [58] found that partner support (i.e., helping with household tasks such as washing dishes, or vacuum cleaning the carpet and child care like changing diapers) resulted in higher rates of EBF and breastfeeding duration for up to the time of weaning.

## 4. Discussion

This review showed evidence which suggests that appropriate partner breastfeeding support is essential for infant feeding and can influence new mothers’ decision to initiate, continue or cease breastfeeding in the early postnatal period. In this review, while there are variations in the types of partner breastfeeding support, the impact of partner support on breastfeeding initiation, duration and exclusivity was largely positive, particularly when the support was provided in the form of verbal encouragement. Other relevant and important types of partner supportive efforts that led to positive breastfeeding behavior included responsiveness of the partner, assistance in preventing and managing breastfeeding difficulties, and helping with household and child care duties. Although this review has shown that the specific types of support of the partner can have a significant impact in the success of breastfeeding in the community, it also demonstrates that there is still a lack of high-quality, population-based studies on the influence of the specific role of partners on breastfeeding behaviors.

Breastfeeding initiation and exclusivity are the cornerstones of appropriate infant and young child feeding, which have benefits not only for the child but also for the mother, the household and the community [2,59,60]. Since the 1980s, the global efforts to protect, promote and support breastfeeding have mainly focused on new mothers and their environments. These efforts include the International Code of Marketing of Breast-milk Substitutes [61]; the Innocenti Declaration [62]; the Baby Friendly Hospital Initiative (BFHI) [63]; and more recently, the Global Nutrition Targets 2025 [11] and Sustainable Development Goals [64]. Our review has shown some evidence that future efforts should not only target new mothers but should also involve their partners and how they can best support new mothers. This is because partners/fathers can play an important role in a mother’s decision to appropriately breastfeed given the hierarchical structure of authority in the households. Additionally, the involvement of grandmothers (if available) in future breastfeeding efforts will also maximise impacts as previous research has shown that grandmothers can also influence breastfeeding behavior in the community [20].

In high-income countries such as Australia and the United Kingdom, breastfeeding support organisations (the Australian Breastfeeding Association [65] and the Association of Breastfeeding Mothers [66], respectively) usually provide information, including the 24 h national breastfeeding helpline in Australia, to support breastfeeding mothers and families in the community. However, these important breastfeeding support organisations are often not available in LMICs, where the benefits of breastfeeding are well recognised [2]. Despite this gap, the World Breastfeeding Trends Initiative ranked LMICs (such as Sri Lanka, Cuba and Bangladesh) higher than high-income countries (like the United States, Germany and Australia) based on 10 indicators on breastfeeding policy and programmes [67]. Improving breastfeeding practices among new mothers would, therefore, require strengthening breastfeeding policies and formulating programmes that also involve household members, who play an important role in the early postpartum period. In recently published systematic reviews, Tadesse et al. [29] and Abbass-Dick et al. [30] found that breastfeeding education that targeted fathers significantly improved breastfeeding practices. In designing breastfeeding programmes, efforts must be made to simplify and provide comprehensive breastfeeding information to fathers, as well as training for those who will deliver the information. This is because past studies have shown that although fathers want to help new mothers to have successful breastfeeding, limited breastfeeding information given to fathers and/or conflicting information from health care workers to fathers were barriers to fathers’ involvement in breastfeeding support [55,58,68].

### Limitations and Strengths

In this review, the eligible studies were limited in number (only seven studies) globally and across time and all study designs, which may not provide sufficient evidence on the impact of specific types of partner support on breastfeeding initiation, duration and exclusivity. Similarly, the review included only studies in the English language and was based on published peer-reviewed studies as relevant data may have been missed from unpublished works. Additionally, this review did not report the pooled estimate of the impact of partner support on EBF from the quantitative studies as the studies estimated varied measure of associations between the specific types of partner support and breastfeeding initiation, duration and exclusivity. The pooled effect from these studies may misrepresent the ‘true’ association between partner support and the breastfeeding outcomes. The cross-cultural generalizability of the study findings is another limitation of this review. Mothers experiences and/or perceptions of the support received from their partners may differ within and between countries. For example, experiences and/or perceptions of mothers regarding the specific types of partner support in high-income countries (e.g., Australia or the United Kingdom) [23,68,69,70,71] may vary from LMICs (e.g., Nigeria [19] or Ghana [72]). Accordingly, future studies and/or breastfeeding programmes should consider culturally-sensitive approaches.

This study has strengths. Firstly, this review has shown that specific types of partner support has the potential to influence the mother’s decision to initiate and continue breastfeeding in the early postnatal period. Secondly, the review of all available studies, regardless of the design, geography and time, provided valuable insights into the impact of partner support on breastfeeding behavior. The inclusion of a qualitative study also offered a holistic explanation for the impact of the specific types of partner supportive actions on breastfeeding initiation, duration and exclusivity [73]. Finally, while the evidence from this review may be limited in scope in relation to the study design, as only two intervention studies were included, it also suggests that future interventional studies that examine the effect of specific types of partner support on breastfeeding are warranted.

## 5. Conclusions

Our study showed that appropriate and specific partner breastfeeding support can influence a mother’s decision to initiate and continue breastfeeding in the early postnatal period. Verbal encouragement to new mothers from their partners was the most common form of support to improve breastfeeding behaviors. Other specific types of partner supportive actions that led to improved breastfeeding behaviors included sensitivity of the partner to the nursing mother’s needs, assistance in preventing and managing breastfeeding difficulties, and helping with household and child care duties. Breastfeeding interventions for new mothers should consider the involvement of partners and their specific roles to maximise impacts.

## Figures and Tables

**Figure 1 ijerph-17-00413-f001:**
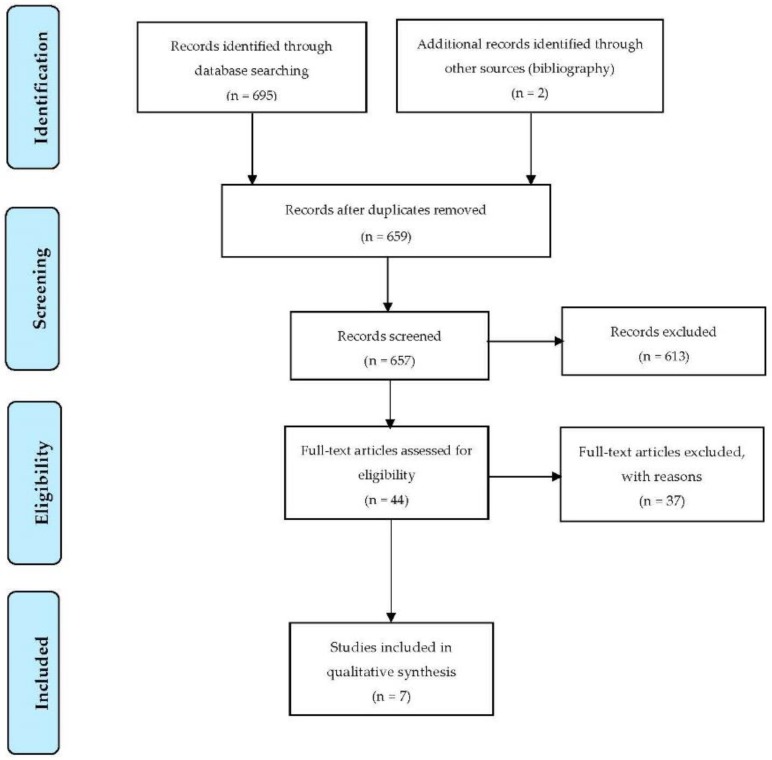
Preferred Reporting Items for Systematic Reviews and Meta-Analyses (PRISMA) flowchart for selection of eligible studies [41].

**Table 1 ijerph-17-00413-t001:** Summary of selected studies.

Author (Year) COUNTRY	Study Design	Study Characteristics	Specific Type of Partner Support	Impact of Partner Support	Study Limitations	Quality Assessment Outcome
Su-Ying Tsai (2014) Taiwan [52]	Cross-sectional questionnaire-based study	Participants: 608 mothers Age: 30–39 years Response rate: 72.9%	Encouragement to use lactation room and milk expression breaks	Increased breastfeeding	(i) Cross-sectional in nature (ii) Dichotomized classification for the assessment of predictors was simplistic (iii) Employed self-report for predictor measurements	Fair
Rempel et al. (2017) Canada [53]	Cross-sectional questionnaire-based study	**Study 1**: Participants: 64 women; 41 men (34 couples) **Study 2**: Participants: 80 mothers; 65 fathers (63 couples)	(i) Responsiveness—father’s sensitivity to the mother’s needs and respect for her decisions (ii) Appreciation—behaviors of encouragement and valuing the breastfeeding mother	(i) Responsiveness led to longer breastfeeding duration (ii) Appreciation led to shorter breastfeeding duration	(i) Cross-sectional in nature (ii) Issue with the stability of correlations due to small samples and large number of correlations (iii) Selection bias	Fair
Ingram et al. (2002) England [54]	Prospective cohort study	Participants: 1174 mothers Mean maternal age: 29.5 years	Giving encouragement	Uptake and continuation of breastfeeding	Failure to control for secular changes in breastfeeding practices over the duration of the study	Fair
Tohotoa et al. (2009) Australia [55]	Qualitative exploratory study	Participants: 48 mothers; 28 fathers Mean age (mothers): 27.5 years Mean age (fathers): 37 years	(i) Anticipating needs and getting the job done (ii) Giving encouragement (iii) Having a paternal commitment to breastfeeding	Successful breastfeeding	(i) Non-representative (ii) Selection bias due to self-selected sample (iii) Small sample size	Fair
Tarrant et al. (2009) Ireland [56]	Cross-sectional questionnaire-based study	Participants: 401 mothers Mean maternal age: 29.5 years	Positive postnatal encouragement to breastfeed	Breast-feeding initiation	(i) Cross-sectional design (ii) Not generalizable (iii) Selection bias	Fair
Pisacane et al. (2005) Italy [57]	Controlled trial	Participants: 280 couples 20% loss to follow-up Mean maternal age: 27.5 years	Assistance with preventing and managing lactation difficulties	Higher rates of exclusive breastfeeding	(i) Limited numbers of participants enrolled (ii) Single hospital setting (iii) Sequential rather than random allocation of the participants	Good
Susin et al. (2008) Brazil [58]	Controlled clinical trial	Participants: 586 families Control group: 201 couples Intervention group A: 192 couples with only mothers exposed to the intervention Intervention group B: 193 couples with mothers and fathers exposed to the intervention	Helping with household tasks and child care	Increased rates of exclusive breastfeeding	(i) Single hospital setting (ii) Sequential rather than random allocation of the participants	Good

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
