# Peer review of "Breastfeeding in the Community—How Can Partners/Fathers Help? A Systematic Review"

_ijerph, 2020, doi:10.3390/ijerph17020413_

Round 1

Reviewer 1 Report

The manuscript,"Breastfeeding in the community- how can partners/fathers help? A systematic review" presents an interesting  approach to the recolect informations about the influence of family members in breastfeedings. It is needed, and useful review of the current status of “data publication” from a certain perspective. The manuscript presents a good and actualized bibliography. The study is of interest for the scientific community.

While the introduction  appears to be sound, the language is unclear, making it difficult to follow. I advise the authors work with a writing coach or copyeditor to improve the flow and readability of the text. The authors are off to a good start, however, this study requires additional datas, particularly, the authors should include more information that clarifies and justifies their choice of articles and the exclusion of them.

Minor points to consider in subsequent versions:

Page 3; line 100-101: the concept of PRISMA could be a bit better explained.

Page 4; line 170-175, I think It is important to specify why the other studies have been excluded

Page 8; The conclusions seem to be descriptive and poor. They should be rewritten in more detail, clearly indicating (citing) the findings found.

Author Response

Comments and Suggestions for Authors

The manuscript,"Breastfeeding in the community- how can partners/fathers help? A systematic review" presents an interesting  approach to the recolect informations about the influence of family members in breastfeedings. It is needed, and useful review of the current status of “data publication” from a certain perspective. The manuscript presents a good and actualized bibliography. The study is of interest for the scientific community.

Response:

We thank the reviewer for the comment.

While the introduction  appears to be sound, the language is unclear, making it difficult to follow. I advise the authors work with a writing coach or copyeditor to improve the flow and readability of the text. The authors are off to a good start, however, this study requires additional datas, particularly, the authors should include more information that clarifies and justifies their choice of articles and the exclusion of them.

Response:

We thank the reviewer for the comment and the concerns raised are addressed below and in the revised manuscript. We, however, note that the other reviewer found our language to be clear, with some minor revisions requested.

Minor points to consider in subsequent versions:

Page 3; line 100-101: the concept of PRISMA could be a bit better explained.

Response:

Point appreciated and now reflected in the revised manuscript (Page 3).

Page 4; line 170-175, I think It is important to specify why the other studies have been excluded

Response

Point appreciated and now reflected in the revised manuscript (Page 4).

Page 8; The conclusions seem to be descriptive and poor. They should be rewritten in more detail, clearly indicating (citing) the findings found.

Response:

We disagree with the reviewer that the conclusion (below) is poor as we summarised the results found instead of just ‘re-writing’ the results, and we offered a receommedation in the concluding section of the manuscript as is the standard pratice in most epidemiological studies.

Our study showed that appropriate and specific partner breastfeeding support can influence a mother’s decision to initiate and continue breastfeeding in the early postnatal period. Verbal encouragement to new mothers from their partners was the most common form of support to improve breastfeeding behaviors. Other specific types of partner supportive actions that led to improved breastfeeding behaviors included sensitivity of the partner to the nursing mother’s needs, assistance in preventing and managing breastfeeding difficulties, and helping with household and child care duties. Breastfeeding interventions for new mothers should consider the involvement of partners and their specific roles to maximise impacts.

Reviewer 2 Report

The objective of the paper is to systematically assess the impact of specific type of partner support on breastfeeding initiation, duration and exclusivity. Following the Preferred Reporting Items for Systematic reviews and Meta-Analysis (PRISMA), 7 studies met the inclusion criteria. The review revealed that supportive actions of partners (verbal, assistance etc.) positively improved breastfeeding practices.

Overall, the research question is important, and the results shed light on potential strategies to maximise the impact of breastfeeding policies and interventions. However, there are some shortcomings regarding the research design and the relation to previous knowledge/studies.

Here are my main comments and areas that needs improvement:

Introduction:

The introduction section includes studies of breastfeeding worldwide, and determinants of EBF and EIBF, whereas studies of partners/father’s/grandmother´s role in a mother´s decision to initiate and continue breastfeeding are mentioned very briefly (22, 23, 26-28). I think that the introduction section can be improved if some more results from these studies were mentioned. In their discussion section, the authors mention two recently published systematic reviews (Tadesse et al. 2018 and Abbass-Dick et al. 2019) of breastfeeding interventions targeted to partners/fathers. I think these systematic reviews should be mentioned in the introduction section, and that the authors relate their review to the published ones. What kind of new information can be found in this review compared to the published ones? What can be the added value of this review?

Sample and Method:

A major question regarding the method corresponds to the first phase of the screening process: exclusion of studies after reading the titles (n=501). I believe it is more common to conduct a systematic review (following the PRISMA flowchart) by first reviewing the abstract, or the abstract together with the title. As a reader of the paper, I wonder what impact this might have had on the numbers of studies finally included in the review. Why did the authors use this strategy? Also, at page 3, line 141 the authors wrote: “The first phase involved the screening of the titles based on suitability with study objective” Which were the criteria used here? Among the 7 studies included in the review, the majority are conducted in high-income countries. How can this affect the results from the systematic review? It could be clearer that only published peer-review articles met the full criteria for inclusion in the study. At page 4, line 115 the authors described that studies were included if they were peer-reviewed articles and dissertations, books and book chapters, working papers etc. Did the authors found any other forms of studies (dissertations etc.) at all? Page 3, line 101: It would be preferred if the authors mention some of the 27 items that were used to ensure the research transparency following the PRISMA statement. Page 3, line 110-111: “An initial search of…… to ensure no previous study on the impact of partner support on exclusive breastfeeding had been conducted”. What do the author mean? A systematic review of partners supports on exclusive breastfeeding? Or any kind of study regarding partners supports on exclusive breastfeeding? Since part of the aim is to study breast feeding initiation, duration and exclusivity, why only searching for previous studies regarding “exclusive breastfeeding”? Page 4: line 168: “Ethical approval was not necessary for this study.” I think that the authors can mention the reason why an ethical approval was not necessary. For example, “By using a systematically review of previous studies, this study……”

Discussion:

Line 304: “The inclusion of a qualitative study also offered alternative explanations”. What kind if alternative explanations did the qualitative study offered?

Line 305-307: “Finally, while the evidence from this review may be limited in scope in relation to the study design…”. What factors in the study design are associated with the limitations?

Author Response

The objective of the paper is to systematically assess the impact of specific type of partner support on breastfeeding initiation, duration and exclusivity. Following the Preferred Reporting Items for Systematic reviews and Meta-Analysis (PRISMA), 7 studies met the inclusion criteria. The review revealed that supportive actions of partners (verbal, assistance etc.) positively improved breastfeeding practices.

Overall, the research question is important, and the results shed light on potential strategies to maximise the impact of breastfeeding policies and interventions. However, there are some shortcomings regarding the research design and the relation to previous knowledge/studies.

Response

We thank the reviewer for the comment and the shortcomings identified are addressed below and in the revised manuscript.

Here are my main comments and areas that needs improvement:

Introduction:

The introduction section includes studies of breastfeeding worldwide, and determinants of EBF and EIBF, whereas studies of partners/father’s/grandmother´s role in a mother´s decision to initiate and continue breastfeeding are mentioned very briefly (22, 23, 26-28). I think that the introduction section can be improved if some more results from these studies were mentioned.

Response

Point appreciated and now reflected in the revised manuscript (Page 2).

In their discussion section, the authors mention two recently published systematic reviews (Tadesse et al. 2018 and Abbass-Dick et al. 2019) of breastfeeding interventions targeted to partners/fathers. I think these systematic reviews should be mentioned in the introduction section, and that the authors relate their review to the published ones. What kind of new information can be found in this review compared to the published ones? What can be the added value of this review?

Response

Point appreciated and now reflected in the revised manuscript (Page 2).

Sample and Method:

A major question regarding the method corresponds to the first phase of the screening process: exclusion of studies after reading the titles (n=501). I believe it is more common to conduct a systematic review (following the PRISMA flowchart) by first reviewing the abstract, or the abstract together with the title. As a reader of the paper, I wonder what impact this might have had on the numbers of studies finally included in the review. Why did the authors use this strategy?

Response

We appreciate the reviewer’s comment. The search strategy was repeated to ensure that no relevant studies were missed, consistent with the PRISMA guidelines and in response to the reviewer comment. The text and the corresponding PRISMA flowchart has been edited in the revised manuscript (Page 4-5).

Also, at page 3, line 141 the authors wrote: “The first phase involved the screening of the titles based on suitability with study objective” Which were the criteria used here?

Response

As noted above, the search was repeated and changes to the text are now reflected in the revised manuscript (Page 4).

Among the 7 studies included in the review, the majority are conducted in high-income countries. How can this affect the results from the systematic review?

Response

Given that the majority of the studies were from high-income countries, our results may be limited in terms of the external valididty. This point was noted in the original manuscript (Limitation section).

The cross-cultural generalizability of the study findings is another limitation of this review. Mothers experiences and/or perceptions of the support received from their partners may differ within and between countries. For example, experiences and/or perceptions of mothers regarding the specific type of partner support in high-income countries (eg, Australia or the United Kingdom) [23, 68-71] may vary from LMICs (eg, Nigeria [19] or Ghana [72]).

It could be clearer that only published peer-review articles met the full criteria for inclusion in the study. At page 4, line 115 the authors described that studies were included if they were peer-reviewed articles and dissertations, books and book chapters, working papers etc. Did the authors found any other forms of studies (dissertations etc.) at all?

Response:

Point appreciated and now reflected in the revised manuscript (Page 4).

 Page 3, line 101: It would be preferred if the authors mention some of the 27 items that were used to ensure the research transparency following the PRISMA statement.

Response:

Point appreciated and now reflected in the revised manuscript (Page 3).

Page 3, line 110-111: “An initial search of…… to ensure no previous study on the impact of partner support on exclusive breastfeeding had been conducted”. What do the author mean? A systematic review of partners supports on exclusive breastfeeding? Or any kind of study regarding partners supports on exclusive breastfeeding? Since part of the aim is to study breast feeding initiation, duration and exclusivity, why only searching for previous studies regarding “exclusive breastfeeding”?

Response

The reviewer has eyes for details – thank you for the observation. The text has been edited in the revised manuscript (Page 3).

Page 4: line 168: “Ethical approval was not necessary for this study.” I think that the authors can mention the reason why an ethical approval was not necessary. For example, “By using a systematically review of previous studies, this study……”

Response

Point appreciated and now reflected in the revised manuscript (Page 4).

Discussion:

Line 304: “The inclusion of a qualitative study also offered alternative explanations”. What kind if alternative explanations did the qualitative study offered?

Response

Point appreciated and now reflected in the revised manuscript (Line 362–64).

Line 305-307: “Finally, while the evidence from this review may be limited in scope in relation to the study design…”. What factors in the study design are associated with the limitations?

Response

Point appreciated and now reflected in the revised manuscript (Line 310–14).

Reviewer 3 Report

this analysis reads as a state of the art analysis, but the results of the analysis are not 'special' or extra ordinary', but rather confirming. I therefore would like to push the authors a little bit further, can they quantify the effect (duration, NNT) and can they compare this effect to other type of interventions like BF groups, or financial incentives ? 

Author Response

This analysis reads as a state of the art analysis, but the results of the analysis are not 'special' or extra ordinary', but rather confirming. I therefore would like to push the authors a little bit further, can they quantify the effect (duration, NNT) and can they compare this effect to other type of interventions like BF groups, or financial incentives ?

Response

We thank the reviewer for the first comment. We, however, note that the reviewer’s comment on the estimation of NNT is inappropriate in this context. The number needed to treat (NNT) is the average number of  patients that need to be treated to prevent one additional adverse outcome (death, stroke, etc.). The NNT is often applicable in clinical trials and an important measure in pharmacoeconomics. No changes made to the manuscript.

Round 2

Reviewer 2 Report

The authors have adequately addressed and carefully revised their article according to the comments and questions raised in the previous round of review. The structure of the paper is now improved and therefore much clearer, mainly because of the revised section regarding the selection of eligible studies (following the PRISMA flowchart).